A de novo assembly of the sweet cherry (Prunus avium cv. Tieton) genome using linked-read sequencing technology

http://orcid.org/0000-0002-5598-2115 Wang Jiawei 1
Liu Weizhen 2 liuweizhen@whut.edu.cn
Zhu Dongzi 1
Zhou Xiang 3
Hong Po 1
Zhao Hongjun 1
Tan Yue 1
Chen Xin 1
Zong Xiaojuan 1
Xu Li 1
Zhang Lisi 1
Wei Hairong 1
Liu Qingzhong 1 qzliu001@126.com
1 Scientific Observation and Experiment Station of Fruits in Huang-huai Area, Ministry of Agriculture, Shandong Institute of Pomology , Taian, Shandong , China
2 School of Computer Science and Technology, Wuhan University of Technology , Wuhan, Hubei , China
3 Key Laboratory of Agricultural Animal Genetics, Breeding, and Reproduction of Ministry of Education, College of Animal Science and Technology, Huazhong Agricultural University , Wuhan, Hubei , China
VanBuren Robert
Electronic publication date: 2020 Jun 5
Publication date: 2020
Volume: 8
Electronic Location ID: e9114
Received 2019 Sep 16; Accepted 2020 Apr 10
Copyright: © 2020 Wang et al.
Copyright year: 2020
Copyright holder: Wang et al.
License: This is an open access article distributed under the terms of the Creative Commons Attribution License, which permits unrestricted use, distribution, reproduction and adaptation in any medium and for any purpose provided that it is properly attributed. For attribution, the original author(s), title, publication source (PeerJ) and either DOI or URL of the article must be cited.
License URL: https://creativecommons.org/licenses/by/4.0/

Keywords: Sweet cherry, Genome sequencing, Genome assembly, 10× Genomics chromium, Linked reads

Funding: Special Fund for Innovation Teams of Fruit Trees in Agricultural Technology System of Shandong Province SDAIT-06-04 Agricultural scientific and technological innovation project of Shandong Academy of Agricultural Science CXGC2018F03 This study was supported by Shandong Provincial Key Laboratory for Fruit Biotechnology Breeding, the Special Fund for Innovation Teams of Fruit Trees in Agricultural Technology System of Shandong Province (SDAIT-06-04), and funded by Agricultural scientific and technological innovation project of Shandong Academy of Agricultural Science (CXGC2018F03). The funders had no role in study design, data collection and analysis, decision to publish, or preparation of the manuscript.

==============================
The sweet cherry (Prunus avium) is one of the most economically important fruit species in the world. However, there is a limited amount of genetic information available for this species, which hinders breeding efforts at a molecular level. We were able to describe a high-quality reference genome assembly and annotation of the diploid sweet cherry (2n = 2x = 16) cv. Tieton using linked-read sequencing technology. We generated over 750 million clean reads, representing 112.63 GB of raw sequencing data. The Supernova assembler produced a more highly-ordered and continuous genome sequence than the current P. avium draft genome, with a contig N50 of 63.65 KB and a scaffold N50 of 2.48 MB. The final scaffold assembly was 280.33 MB in length, representing 82.12% of the estimated Tieton genome. Eight chromosome-scale pseudomolecules were constructed, completing a 214 MB sequence of the final scaffold assembly. De novo, homology-based, and RNA-seq methods were used together to predict 30,975 protein-coding loci. 98.39% of core eukaryotic genes and 97.43% of single copy orthologues were identified in the embryo plant, indicating the completeness of the assembly. Linked-read sequencing technology was effective in constructing a high-quality reference genome of the sweet cherry, which will benefit the molecular breeding and cultivar identification in this species.

Introduction

The sweet cherry (Prunus avium), originated in Asia Minor near the Black Sea and the Caspian Sea. It is known as one of the most economically significant fruit species in the world (Quero-García et al., 2017) and its production in China has increased dramatically over the last three decades with the expansion of acreage dedicated to its cultivation. Recent breeding efforts have focused on improving yield, fruit quality, tree architecture and biotic and abiotic resistance (Aranzana et al., 2019). Sweet cherry and other Prunus crops have a long juvenile period, which means that traditional breeding methods are slow to produce improvements (Quero-García et al., 2017). Marker-assisted breeding and genomic selection can speed up the breeding cycle, but these methods require a high-quality reference genome in order to obtain a sufficient amount of genetic variants and to identify the regulatory regions controlling the morphological and physiological characteristics of the plant (Aranzana et al., 2019; Ru et al., 2015). Only one draft genome assembly of sweet cherry cv. Satonishiki (Shirasawa et al., 2017) and one mitochondrial genome sequence of cv. Summit have been reported (Yan et al., 2019), despite the simple genome of the sweet cherry (2n = 2x = 16). The draft genome of sweet cherry cv. Satonishiki was sequenced using Illumina short-read sequencing technology, resulting in a fragmented assembly of 272.4 MB with a scaffold N50 of 219.6 KB (Shirasawa et al., 2017). The linked-read sequencing pipeline developed by 10× Genomics may result in more continuous genomes for the sweet cherry at a lower financial cost (Pollard et al., 2018; Zheng et al., 2016). This technology use a barcoded sequencing library to generate long-range information (preferably >100 KB) and standard short-read sequencing to ensure massive throughput and high accuracy. It was designed for human genome assembly, but has been used effectively in many other animal and plant species, including the wild dog, proso millet pepper and soybean (Armstrong et al., 2018; Hulse-Kemp et al., 2018; Liu et al., 2018; Ott et al., 2018).

We demonstrated that linked-read technology is effective in the de novo assembly of the genome of the sweet cheery cv. Tieton, which is the most popular cherry variety in China. The sweet cherry cv. Tieton genome assembly surpasses the cv. Satonishiki genome assembled using Illumina short-reads in continuity, with a tenfold improvement of scaffold N50 (Shirasawa et al., 2017). The high-quality genome assembly and annotation in this study are valuable for genetic marker development and gene mapping, which may improve sweet cherry breeding. Our assembly platform will support future de novo genome assemblies for other Prunus crops using the linked reads method.

Materials and Methods

Sample and DNA extraction

Leaf samples were collected from the sweet cherry cv. Tieton grown in the experimental orchard of Shandong Institute of Pomology, Taian, Shandong Province, China, and frozen in liquid nitrogen. High-molecular-weight (HMW) genomic DNA (gDNA) was extracted from the frozen leaves using MagAttract HMW DNA Kit (Qiagen, Hilden, Germany) following the manufacturer’s protocol. The gDNA was quantified using Implen NanoPhotometer P330 (Implen, Munich, Germany) and assessed using agarose gel electrophoresis.

Chromium library construction and sequencing

The single Chromium library was constructed by CapitalBio Technology Inc. (Beijing, China) using the purified HMW gDNA sample. The library was sequenced in one lane as 150 nt-Chromium-linked paired-end reads on an Illumina HiSeq X Ten sequencer (Illumina, http://www.illumina.com/). We filtered out raw reads with >5% undetermined bases (Ns), >30% nucleotides quality score lower than 20, and the adapter sequence overlap >5 bp.

De novo assembly and evaluation

We estimated the size of the sweet cherry genome based on the k-mer frequency of the sequence data using the k-mer counting program Jellyfish (v.2.0.8) (Marcais & Kingsford, 2011) and GenomeScope (v1.0.0) (Vurture et al., 2017). The genome was assembled and scaffolded using the Supernova assembler (v2.0, https://www.10xgenomics.com/). This program links sequencing reads to the originating HMW DNA molecule using barcoded information and constructs phased, whole-genome de novo assemblies form the Chromium-prepared library (Weisenfeld et al., 2017). Chromium-linked reads of different sizes (40×, 50×, 60×, 65×, 68×, 70× and 75×) were used as input data. The assembly, using 70× coverage of the reads, was selected for analysis based on superior quality, and higher contig N50 and scaffold N50. Default parameters were set and two pseudohap assemblies were generated; pseudohap1 was used for further analysis. A total of 150 million reads were sampled and aligned to the assembled genome sequence; the quality of the sweet cherry cv. Tieton genome assembly was evaluated using the Burrows–Wheller Alignment tool (BWA, 0.7.17-r1188) (Li & Durbin, 2009). Core Eukaryotic Genes Mapping Approach (CEGMA, v2.5) (Parra, Bradnam & Korf, 2007) and Benchmarking Universal Single-Copy Orthologs (BUSCO, v3.0, embryophyta_odb10) (Simao et al., 2015) were used to assess the completeness of the assembly.

Chromosome-scale pseudomolecule construction

Scaffolds were assembled using the Supernova assembler and were ordered and oriented using seven previously published sweet cherry genetic maps for the construction of the chromosome-scale pseudomolecules. Five of the seven maps were built by Shirasawa et al. (2017), Peace et al. (2012), Klagges et al. (2013), Calle et al. (2018) and Guajardo et al. (2015). The initials of the first author were used to name their respective maps and the maps are referred to as KS, CP, CK, AC and VG. The other two maps, named JWF (the framework map of the WxL map) and JWF1 (the second round of the WxL map), were both reported by Wang et al. (2015). Genetic markers and/or flanking sequences for these maps were aligned to the current scaffolds using GMAP (v2018-07-04) (Wu & Watanabe, 2005) as described by Hulse-Kemp et al. (2018). Markers were manually filtered out if they were aligned to more than one scaffold or the same scaffold in different linkage groups. The alignment results of GMAP were fitted into ALLMAPS (v0.8.4)(Tang et al., 2015) to generate the final consensus map and chromosome-scale pseudomolecules. Different weight parameters were tried for the seven linkage maps and the optimal weight settings with the largest number of anchored and oriented scaffolds were: KS = 2, CP = 3, CK = 1, AC = 1, VG = 1, JWF = 1 and JWF1 = 1.

Identification of repetitive elements in sweet cherry genome

Homology-based and de novo methods were combined to identify repetitive and transposon elements in our final assembly using RepeatMasker (v.4.0.6) (Smit, Hubley & Green, 2016) and RepeatModeler (v.1.0.11) (http://www.repeatmasker.org/RepeatModeler.html).

RNA-seq analysis

Total RNA was extracted from the young leaves of a single plant for genome sequencing. The cDNA library was constructed based on the description of Wei et al. (2015) and sequenced by CapitalBio Technology Inc. (Beijing, China) using the Illumina HiSeq 2000 platform. The adapters were trimmed and low-quality reads were removed before the remaining high quality reads were assembled by Trinity (v2.8.5) (Grabherr et al., 2011).

Non-coding RNA prediction, protein-coding gene prediction and functional annotation

INFERNAL (v1.1.2) (Nawrocki, Kolbe & Eddy, 2009) was used to identify the non-coding RNAs (ncRNAs) in the sweet cherry cv. Tieton genome against the RFAM database (Griffiths-Jones et al., 2005). The tRNAs were identified by tRNAscan-SE (v2.0.5) (Lowe & Eddy, 1997). The rRNAs were identified using RNAmmer (v1.1.2) (Lagesen et al., 2007).

Homology-based, de novo and RNA-seq methods were combined to predict the protein-coding genes in sweet cherry cv. Tieton genome. Augustus (v3.3.2) (Keller et al., 2011) and SNAP (v2013-11-29) (Korf, 2004) were used in the de novo annotation to predict the protein-coding gene in repeat-masked genome sequences. The predicted genes were annotated by Genewise (v2.4.1) (Birney, Clamp & Durbin, 2004) and Exonerate (v2.4.0) (Slater & Birney, 2005). The Program to Assemble Spliced Alignments (PASA, v2.4.1) pipeline (Haas et al., 2003) was used in transcriptome-assistant method with the unigenes assembled by the RNA-seq data. EVidenceModeler (EVM, v1.1.1) (Haas et al., 2008) and PASA were used to combine the predicted results.

Gene family analysis

OrthoFinder (v2.2.7) (Emms & Kelly, 2015) was used to identify the orthologous genes from 13 plant genomes of the sweet cherry cv. Tieton (Prunus avium, Pa), peach (Prunus persica, Pp), Chinese plum (Prunus mume, Pm), flowering cherry (Prunus yedoensis, Py), apple (Malus x domestica, Md), pear (Pyrus bretschneideri, Pb), black raspberry (Rubus occidentalis, Ro), strawberry (Fragaria vesca, Fv), rose (Rosa chinensis, Rc), orange (Citrus sinensis, Cs), grape (Vitis vinifera, Vv), tomato (Solanum lyconpersicum, Sl), and arabidopsis (Arabidopsis thaliana, At) (The Tomato Genome Consortium, 2012; Zhang et al., 2012; Wu et al., 2013; Xu et al., 2013; Canaguier et al., 2017; Daccord et al., 2017; Li et al., 2017; Verde et al., 2017; Baek et al., 2018; Raymond et al., 2018; Sloan, Wu & Sharbrough, 2018; VanBuren et al., 2018). The protein sequences of each plant genome were generated from their most recently annotated versions and were used as input sequences for OrthoFinder. Table S1 shows the annotated version and reference of the other 12 plant genomes except for our sweet cherry cv. Tieton genome. CAFÉ (v4.2) (De Bie et al., 2006) was used to analyze the expansion and contraction of their gene families. The species tree was generated using STRIDE (Emms & Kelly, 2017), as part of OrthoFinder and used as the input phylogenetic tree for CAFÉ.

Comparison between sweet cherry cv. Tieton genome and cv. Satonishiki genome

D-GENIES (v1.2.0) was used to compare the sweet cherry cv. Tieton genome with the cv. Satonishiki genome (Cabanettes & Klopp, 2018; Shirasawa et al., 2017). The whole sequence synteny analysis of the two assemblies were compared in both scaffold level and pseudochromosome level.

To compare the gene content between the two genome assemblies, we used three annotation versions that are the sweet cherry cv. Tieton genome annotation, the cv. Satonishiki genome annotation (Shirasawa et al., 2017), and an improved and re-annotated assembly of cv. Satonishiki genome released by NCBI Eukaryotic Genome Annotation Pipeline (NCBI Prunus avium Annotation Release 100, https://www.ncbi.nlm.nih.gov/genome/annotation_euk/Prunus_avium/100/). OrthoFinder was used to compare the gene content among the three annotations (Emms & Kelly, 2015).

Results and Discussion

Sequencing summary

For sweet cherry cv. Tieton, a total of 121.61 GB of raw sequencing data was generated with more than 810 million Chromium-linked paired-end reads. Table 1 shows the statistics of the sequencing for the linked-read library. The low quality reads were filtered out and 750,890,534 clean reads were used for de novo assembly. The average Q20 was 97.52% and GC content was 40.8%. A cDNA library was constructed and sequenced to improve the precision of the genome annotation. As shown in Table S2, over 78 million 150-nt length paired-end reads were generated and assembled.

Table 1 Raw data and valid data statistics of sequencing for linked-read libraries of sweet cherry (Prunus avium) cv. Tieton.

Parameter	Value	Parameter	Value	
Raw bases (Gb)	121.61	Clean bases (Gb)	112.63	
Q20 (%)	97.52	Clean reads	750,890,534	
Q30 (%)	94.24	Clean ratio (%)	92.62	
GC content (%)	40.8	Low ratio (%)	5.51	
N ratio (%)	0.01	Adapter ratio (%)	1.86	

Determination of genome size and heterozygosity

The genome size of sweet cherry cv. Tieton was estimated to be 341.38 MB based on 37-nt k-mer, which is very close to the genome size of 338 MB estimated by flow cytometry (Arumuganathan & Earle, 1991). The k-mer distribution generated by GenomeScope was shown in Fig. S1. The sweet cherry cv. Satonishiki genome estimated by k-mer method was 352.9 MB (Shirasawa et al., 2017), larger than cv. Tieton genome. The genome size difference is probably due to the variety difference, but also may be caused by different library construction and sequencing methods. Heterozygosity of sweet cherry cv. Tieton genome was estimated to be 0.45%, and the repeat content was estimated to be 48.5% as shown in Fig. S1.

Genome assembly and quality-assessment

The Supernova assembler (version 2.0) was used in de novo assembly and different sizes (40×, 50×, 60×, 65×, 68×, 70×, and 75×) of the Chromium-linked reads were attempted (Weisenfeld et al., 2017). Table S3 listed these assembly results, illustrating that the assembly using 70× coverage reads has the best assembly quality, and was selected for following analyses. GapCloser filled gaps in the raw sequencing data (Luo et al., 2012), resulting in the draft genome assembly of sweet cherry cv. Tieton of 280.33 MB with contig N50 and scaffold N50 sizes of 63.65 KB and 2.48 MB, respectively. Our sweet cherry cv. Tieton genome assembly had tenfold better contiguity than the cv. Satonishiki genome assembly (Shirasawa et al., 2017). The whole assembly increased in size from 272.36 to 280.33 MB, whereas scaffold N50 increased from 219 KB to 2.48 MB (Table 2).

Table 2 Comparison of sweet cherry (Prunus avium) genome assemblies of cv. Tieton and cv. Satonishiki.

Assembly parameters	cv. Tieton	cv. Satonishiki	
Assembled genome size (Mb)	280.33	272.36	
Scaffold N50 (Mb)	2.48	0.22	
Number of scaffold	14,344	10,148	
Longest of scaffold (Mb)	17.96	1.46	
Contig N50 (kb)	63.65	28.779	
Number of contig	19,420	32,301	
Longest of contig (kb)	670.29	19.97	
Total contig length (Mb)	237.92	246.8	
GC content (%)	37.86	37.7	
Ns (%)	15.12	9.34	
Note:

Mb, Megabase; kb, Kilobase; GC, Guanine-cytosine; Ns, Ambiguous bases.

A total of 150 million reads were sampled and 99.02% of the sampled reads were aligned to the sweet cherry cv. Tieton genome sequence using BWA (Li & Durbin, 2009), shown in Table S4. CEGMA (Parra, Bradnam & Korf, 2007) and BUSCO (Simao et al., 2015) were used to evaluate the completeness of the sweet cherry cv. Tieton genome and results were summarized in Table S5. Out of 248 core eukaryotic genes, 231 and 13 were found to be complete and partial genes in the CEGMA assessment, respectively. BUSCO analysis showed that our assembly captured 1,403 (97.43%) of the 1,440 single-copy orthologous genes of the embryo plant, of which 1,381 (95.9%) were complete (1,345 single-copy and 36 duplicated-copy), showing that the sweet cherry cv. Tieton genome assembly is well covered the gene space of the sweet cherry genome.

Chromosome-scale pseudomolecule construction

A consensus map was constructed from previously reported sweet cherry genetic maps for the chromosome-scale pseudomolecule construction (Calle et al., 2018; Guajardo et al., 2015; Klagges et al., 2013; Peace et al., 2012; Shirasawa et al., 2017; Wang et al., 2015). GMAP (Wu & Watanabe, 2005) and ALLMAPS (Tang et al., 2015) were used to organize scaffolds onto eight chromosome-scale pseudomolecules (Hulse-Kemp et al., 2018). A total of 494 scaffolds representing more than 214 MB sequences, were anchored to eight chromosome-scale pseudomolecules of the sweet cherry cv. Tieton genome using 7,838 genetic markers (36.6 markers per Mb). 202.6 of the 214 MB anchored sequences were oriented, the anchor rate and synteny of the maps were shown in Table S6 and Fig. 1. This formation resulted in a higher contiguity than the sweet cherry cv. Satonishiki genome, consisting of 905 scaffolds spanning 191.7 MB (Shirasawa et al., 2017).

Figure 1 Pseudomolecule construction of sweet cherry (Prunus avium) by assigning scaffolds to seven genetic maps.

Chr 1–8 represents constructed pseudomolecules by merging seven genetic maps. AC, VG, CK, CP, KS, JWF, and JWF1 denote the sweet cherry genetic maps reported in Calle et al. (2018), Guajardo et al. (2015), Klagges et al. (2013), Peace et al. (2012), Shirasawa et al. (2017) and Wang et al. (2015), respectively.

Annotation of repetitive sequences

The Repbase library and repetitive motifs were searched and 32.71% (over 91 MB) of the sweet cherry cv. Tieton genome assembly was found to be repetitive. Different repetitive elements were annotated in sweet cherry cv. Tieton genome, and their distribution were shown in Table 3. Long-terminal-repeat retrotransposons (6.39%) were predominant among the repetitive elements. The annotated repeat sequence length of the sweet cherry cv. Tieton genome was 28.4 MB shorter than the sweet cherry cv. Satonishiki genome (Shirasawa et al., 2017), which may explain why the k-mer method estimated a smaller genome size for cv. Tieton than cv. Satonishiki (299.17 vs. 352.9 MB).

Table 3 Summary of detected repeat elements of sweet cherry (Prunus avium) cv. Tieton genome.

Repeat type	Number	Total length (bp)	Percent (%)	
LTR	22,244	17,899,535	6.39	
DNA elements	11,927	7,198,678	2.57	
LINE	4,700	1,900,833	0.68	
SINE	1	84	0	
Simple repeat	6,266	4,736,127	1.69	
Low complexity	141	23,252	0.01	
Unknown	228,932	59,943,002	21.38	
Total	274,211	91,701,511	32.71	
Note:

LTR, Long terminal retrotransposon; SINE, Short interspersed nuclear elements; LINE, Long interspersed nuclear elements.

cDNA assembly and noncoding RNA (ncRNA) annotation

Trinity was used to assembly the high quality cDNA reads (Grabherr et al., 2011). A total of 33,401 transcripts with a total length of 42.6 MB were generated. The length of the assembled transcripts ranged from 201 to 15,591 nt, with a mean length of 1,276 nt. These assembled contigs were considered to be unigenes, and the distribution of their lengths is shown in Table S7.

Noncoding RNA includes miRNA, rRNA, snoRNA, tRNA, and the tRNA pseudogene. A total of 109,277 ncRNAs were generated, with a total length of 7.35 MB, representing 2.63% of the sweet cherry cv. Tieton genome. As summarized in Table 4, our annotation predicted fewer tRNAs and rRNAs, compared to the annotation in of sweet cherry cv. Satonishiki genome (Shirasawa et al., 2017).

Table 4 Summary of noncoding-RNAs prediction in sweet cherry (Prunus avium) cv. Tieton genome.

Non-coding RNA type	Non-coding RNA number	Total length (bp)	Percentage (%)	
miRNA	21,673	1,703,848	0.61	
rRNA	35	51,780	0.02	
snoRNA	86,993	5,560,365	1.98	
tRNA	521	39,227	0.01	
tRNA-pseudogene	48	3,585	0	
Total	109,277	7,358,805	2.63	
Notes:

miRNA, micro-RNA; rRNA, ribosomal RNA; snoRNA, small nucleolar RNA; tRNA, transfer RNA

Percentage(%)=thetotallengthofcorrespondingnon−codingRNAtypewholegenomesizeofcv.Tieton.

Protein-coding gene prediction and functional annotation

In total, 30,439 genes coding for 30,975 proteins were predicted in the sweet cherry cv. Tieton genome assembly. A summary of the predicted results using different methods was shown in Table 5. The de novo methods predicted 47,866 gene models, but the average gene length was shorter than other methods. After correcting with the transcript evidence, more than 16,000 genes were filtered out.

Table 5 Statistics for protein-coding gene prediction of sweet cherry (Prunus avium) cv. Tieton genome.

Prediction method or software	Number of genes	mRNA number	Average RNA length	Exon number	Average exon length	Intron number	Average intron length	
De novo	47,866	47,866	2118.8	179,067	302.9	131,201	359.5	
RNA-seq	16,512	16,512	4032.3	91,646	228.5	75,134	344.6	
EVM	30,455	30,455	2433.3	139,225	275.8	108,770	328.3	
PASA	30,439	30,975	2720.6	140,185	277	109,210	329.2	
Note:

EVM, EVidenceModeler; PASA, Program to Assemble Spliced Alignments.

The predicted 30,975 proteins were blasted against non-redundant protein sequences (NR, https://blast.ncbi.nlm.nih.gov), Uniprot (The UniProt 2017), Kyoto Encyclopedia of Genes and Genomes (KEGG) (Kanehisa et al., 2014), and InterPro (Finn et al., 2017) by using BLASTP (v2.9.0) (Camacho et al., 2009). As shown in Table 6, 30,973 of 30,975 proteins (99.99%) were annotated in at least one database.

Table 6 Statistics of functional annotated genes of sweet cherry (Prunus avium) cv. Tieton genome.

Functional database	Number of annotated genes	Percentage (%)	
InterPro	30,300	97.8	
NR	30,882	99.7	
GO	16,433	53.05	
Uniprot	29,444	95.05	
KEGG	9,202	29.7	
Total	30,973	99.99	
Note:

NR, NCBI Non-redundant protein; GO, Gene ontology; KEGG, Kyoto Encyclopedia of Genes and Genomes.

Gene family analysis compared with other plant species

OrthoFinder (Emms & Kelly, 2015) identified the potential orthologous genes between the sweet cherry cv. Tieton genome and the other 12 plant genomes. The results of gene orthologous analysis were shown in Table S8. Gene family clustering identified 23,129 common orthogroups consisting of 375,493 genes (81.1% of the total genes) in these genomes. 8,465 orthogroups were present in all species, and 246 were single-copy genes. In the sweet cherry cv. Tieton genome, 46 orthogroups (124 genes) were unique and 2,062 orphan genes were identified that could not be clustered with any genes in the thirteen genomes. A species tree was constructed using STRIDE (Emms & Kelly, 2017), as part of OrthoFinder. As shown in Fig. 2, sweet cherry (Prunus avium) exhibits a closer relationship with flowering cherry (Prunus yedoensis) than peach (Prunus persica) and Chinese plum (Prunus mume). A comparison was conducted to evaluate the expansion or contraction of these gene families using CAFÉ (version 4.2) (De Bie et al., 2006), and the results were shown in Fig. 2. A total of 1,012 gene families expanded and 3,642 gene families contracted in the sweet cherry cv. Tieton genome compared to the other 12 plant genomes (Fig. 2).

Figure 2 Species tree and gene family expansion analysis of 13 plant species.

The species tree were constructed using STRIDE. Gene family expansions are indicated in red, and gene family contractions are indicated in green.

Comparison between sweet cherry cv. Tieton genome and cv. Satonishiki genome

According to Fig. 3A, genomic analysis using D-GENIES showed a high scaffold-level synteny of the sweet cherry cv. Tieton genome compared to sweet cherry cv. Satonishiki genome. High chromosome-level synteny was also detected in the two sets of pseudomolecules, except at the end of chromosomes 1, 4, 5, and 6 (Fig. 3B). Based on Fig. 3A, the sweet cherry cv. Tieton genome assembly had a better contig contiguity, whereas the sweet cherry cv. Satonishiki genome was more fragmented.

Figure 3 Synteny analysis between sweet cherry (Prunus avium) cv. Tieton genome and cv. Satonishiki genome.

(A) Scaffold level synteny dot plot. (B) Chromosome-scale synteny dot plot. Sequence identity is indicated by colors.

The original annotation of sweet cherry cv. Satonishiki genome (Shirasawa et al., 2017) and the re-annotated version of cv. Satonishiki genome released by the NCBI Eukaryotic Genome Annotation Pipeline were used to compare the gene content with our annotation of sweet cherry cv. Tieton genome. OrthoFinder analysis showed that the originally annotated version of cv. Satonishiki had 48 species-specific orthogroups represented 349 genes from our cv. Tieton genome annotation and the NCBI annotation of cv. Satonishiki genome (Table 7). The original version of sweet cherry cv. Satonishiki assembly annotated 41% more genes than our cv. Tieton genome annotation, however, the re-annotated version of cv. Satonishiki genome annotated a similar number of genes with our cv. Tieton genome. The increased gene numbers in the original annotation of sweet cherry cv. Satonishiki genome can be attributed to the fragmentation of genes onto multiple individual contigs. The re-annotated version of sweet cherry cv. Satonishiki genome adopted RNA-seq to improve the quality of the gene annotation by connecting genes fragmented in the assembly process (Denton et al., 2014). This method was also used in our sweet cherry cv. Tieton genome annotation process.

Table 7 Statistics of orthogroups analysis between sweet cherry (Prunus avium) cv. Tieton and cv. Satonishiki genome annotations.

Annotation summary	cv. Tieton	cv. Satonishiki	
NCBI version	Original version	
Number of genes	30,975	35,009	43,673	
Number of genes in orthogroups	26,730	31,314	25,388	
Number of unassigned genes	4,245	3,695	18,285	
Percentage of genes in orthogroups	86.3%	89.4%	58.1%	
Percentage of unassigned genes	13.7%	10.6%	41.9%	
Number of orthogroups containing species	21511	21258	20738	
Percentage of orthogroups containing species	92.4%	91.3%	89%	
Number of species-specific orthogroups	14	1	48	
Number of genes in species-specific orthogroups	67	2	349	
Notes:

NCBI version is the improved assembly annotation of sweet cherry cv. Satonishiki released by National Center for Biotechnology Information (https://www.ncbi.nlm.nih.gov/genome/annotation_euk/Prunus_avium/100/).

Original version is the assembly annotation of sweet cherry cv. Satonishiki genome documented in (Shirasawa et al., 2017).

Conclusion

We successfully assembled a high-quality reference genome of sweet cherry cv. Tieton using linked reads sequencing technology. The assembly will be a valuable resource for future breeding efforts, gene function characterization and cultivar identification in the sweet cherry, as well as for comparative genomic analysis with other Prunus species.

Supplemental Information

Supplemental Information 1 Genome size estimation of sweet cherry (Prunus avium) cv. Tieton based on k-mer (37-mer) analysis.

Click here for additional data file.

Supplemental Information 2 Genome annotations used for gene orthologous analysis in this study.

Click here for additional data file.

Supplemental Information 3 Statistics of RNA sequencing of sweet cherry (Prunus avium) cv. Tieton.

Click here for additional data file.

Supplemental Information 4 Statistics of sweet cherry (Prunus avium) cv. Tieton genome assembly using Supernova v2.0 with 40x, 50x, 60x, 65x, 68x, 70x, and 75x coverage of linked reads.

Click here for additional data file.

Supplemental Information 5 Summary of 150 Million reads mapping against sweet cherry (Prunus avium) cv. Tieton genome assembly using Burrows-Wheller Alignment (BWA) tool.

Click here for additional data file.

Supplemental Information 6 Summary of sweet cherry (Prunus avium) cv. Tieton genome completeness assessed by Core Eukaryotic Genes Mapping Approach (CEGMA) and Benchmarking Universal Single-Copy Orthologs (BUSCO).

Click here for additional data file.

Supplemental Information 7 Summary of scaffolds anchored to pseudo-chromosomes of sweet cherry (Prunus avium) cv. Tieton genome.

Click here for additional data file.

Supplemental Information 8 Distribution of the RNA-seq assembly length by Trinity.

Click here for additional data file.

Supplemental Information 9 Statistics of gene family analysis between sweet cherry (Prunus avium) cv. Tieton and the other 12 plant species.

Click here for additional data file.

Additional Information and Declarations

Competing Interests

Author Contributions

Data Availability

The authors declare that they have no competing interests.

Jiawei Wang conceived and designed the experiments, performed the experiments, analyzed the data, prepared figures and/or tables, authored or reviewed drafts of the paper, and approved the final draft.

Weizhen Liu conceived and designed the experiments, analyzed the data, prepared figures and/or tables, authored or reviewed drafts of the paper, and approved the final draft.

Dongzi Zhu conceived and designed the experiments, performed the experiments, analyzed the data, prepared figures and/or tables, authored or reviewed drafts of the paper, and approved the final draft.

Xiang Zhou analyzed the data, authored or reviewed drafts of the paper, and approved the final draft.

Po Hong performed the experiments, analyzed the data, prepared figures and/or tables, authored or reviewed drafts of the paper, and approved the final draft.

Hongjun Zhao performed the experiments, authored or reviewed drafts of the paper, and approved the final draft.

Yue Tan analyzed the data, authored or reviewed drafts of the paper, and approved the final draft.

Xin Chen performed the experiments, authored or reviewed drafts of the paper, and approved the final draft.

Xiaojuan Zong performed the experiments, authored or reviewed drafts of the paper, and approved the final draft.

Li Xu analyzed the data, authored or reviewed drafts of the paper, and approved the final draft.

Lisi Zhang performed the experiments, authored or reviewed drafts of the paper, and approved the final draft.

Hairong Wei analyzed the data, authored or reviewed drafts of the paper, and approved the final draft.

Qingzhong Liu conceived and designed the experiments, authored or reviewed drafts of the paper, and approved the final draft.

The following information was supplied regarding data availability:

Raw sequencing reads are available in GenBank under Bioproject ID PRJNA503752.

Genome assembly, annotation and chromosome-scale pseudomolecule construction date are available at Figshare: Wang, Jiawei; Liu, Weizhen; Zhu, Dongzi; Liu, Qingzhong (2020): A de novo assembly of the sweet cherry (Prunus avium cv. Tieton) genome using linked-read sequencing technology. figshare. Dataset. DOI 10.6084/m9.figshare.9810236.v1. The genome assembly (whole genome shotgun sequencing project) is available at NCBI: VTVB00000000.

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
