# Peer review of "A de novo assembly of the sweet cherry (Prunus avium cv. Tieton) genome using linked-read sequencing technology"

_PeerJ, doi:10.7717/peerj.9114_

## Round 0.1 · original submission · Major Revisions

Your manuscript has been seen by two qualified reviewers. Although the reviewers found value in the genomic resources presented here, they raised some concerns and had suggestions to improve the manuscript.
I have summarized the main concerns that should be addressed if a revised manuscript is submitted:

1. More details are needed on the methods for assembling and anchoring the genome.

2. It would be useful to compare the reference genome presented here to the first sweet cherry genome sequence reported by Shirasawa et al., 2017. Simple comparative genomics analyses and a comparison of gene content differences would help strengthen this manuscript and validate the findings.

3. There are a number of grammatical issues and the manuscript needs some heavy editing.

Reviewer 1 ·

Basic reporting

This paper reports the whole-genome sequence of sweet cherry, for which the authors used 10X Genomics Chromium technology.

Since the assembly presented in this paper is not dramatically improved in compare to the previous report, I recommend to delete "improves genome coverage and completeness" from the title.

Experimental design

The authors used 10X Genomics Chromium reads to estimate the size of the sweet cherry genome (299.17 Mb) and to evaluate the assembly qualith (99.02%). However, as the authors have already pointed out, it might be due to bias in the Chromium library missing 38 Mb. I strongly recommend the authors to analyze again with whole-genome shotgun reads obtained from a PCR-free method, e.g., TruSeq DNA PCR-Free Sample.

Validity of the findings

The text is totally descriptive and lacks any insights into biological aspects in sweet cherry.

Additional comments

Please consider the comments to improve this manuscript.

Reviewer 2 ·

Basic reporting

The paper: A new de novo assembly of sweet cherry (Prunus avium) improves genome coverage and completeness by the authors: Jiawei Wang, Weizhen Liu, Dongzi Zhu, Xiang Zhou, Po Hong, Hongjun Zhao, Yue Tan, Xin Chen, Xiaojuan Zong, Li Xu, Lisi Zhang, Hairong Wei and Qingzhong Liu prsent data for a new sweet cherry sequencing using the "Tieton" variety. I belive that the authors need to improve the introduction of the manuscript. There are few paper that should be considered, for instance: 1) Prunus genetics and applications after denovo genome sequencing: achievements and prospects by Aranzana et al. Horticulture Research ( 2019)6 :58; Also is important to add info about organel sequencing: The complete mitochondrial genome sequence of sweet cherry (Prunus avium
cv. ‘summit’) by Yang et aL., MITOCHONDRIAL DNA PART B 2019, VOL. 4, NO. 1, 1996–1997 AMONG OTHERS. The figures and tables need to be adreess in a more wide aspect.

Experimental design

The paper present a new sweet cherry genome sequencing using two powerful tools that allow to improve the sequencing, however no much information is given for those methods: linked-read sequencing technology and the Supernova genome assembler. These tewchniques are very important for genome sequencing in plants and should be highlighted.

The research question is well defined but I stated previously it need to considerer more publication on the field.

Validity of the findings

The results are good however they should discuss and compare their results with the previos seet cherry sequencing (Shirasawa et al., 2017). This will allow to present more robust conclusions for this sequencing. Most of the genome sequencing papers in these do a deep genome sequencing wth the same species or relates, for instance: peach; sweet cherry and apple.

Anoter concern is why the do not use the supplementary data through the manuscript?

Additional comments

The english need some improvement in the manuscript and supplementary data.

---

## Round 0.2 · Minor Revisions

Your revised manuscript has been seen by one of the original reviewers. Although they feel the manuscript is improved, they identified a number of additional issues that must be addressed in another revision. Most importantly, you should carefully review all of the figures, tables, and raw values in the manuscript to ensure they are consistent and correct.

Reviewer 2 ·

Basic reporting

The authors submitted a new version with a new tittle: “A de novo assembly of the sweet cherry (Prunus avium cv. Tieton) genome using linked-read sequencing technology”. I would like to thanks the authors because in this new version they considered the majority of my concerns nevertheless still they are some issues with the presentation of tables and figures.
The authors need to expand their arguments in the presentation of figures and tables otherwise when they refer to a table and/or figure in the manuscript is not clear what have to be seeing.
Lane 280: “Genomic analysis using D-GENIES showed a high contig-level synteny of the Tieton genome compared to the Satonishiki genome before 196.2 Mb”. The author should review all tables and figures. All of them should say: ….of the sweet cherry cv Tieton genome compared to the sweet cherry cv Satonishiki genome….” Tieton and Satonishiki are varieties, no species. In the same lane it is difficult to understand the meaning of “before 196.2 Mb”.
Table 1, tittle: “Summary statistics of sequence data”. A tittle in a Table has to be self-explanatory. In this case: What sequence data? Of what? Then in the column of samples, How the error date can be a sample? The tables need to be more concise.
Table 2, tittle: “Comparison of sweet cherry (Prunus avium) genome assembly between cv. Tieton in the current study and cv. Satonishiki in previous study”. Should be: “Comparison of sweet cherry (Prunus avium) genome assembly cv. Tieton and cv. Satonishiki” (the authors already gave info about both genomes). The in the foot table explain the abbreviation used!
Table 3, tittle: “Distribution of repeats and unique sequences”. It not clear the tittle. What is the meaning of LINE, SINE?? Why is a – in the table?
Table 4, tittle: “Summary of the none-coding RNA analysis”. None-coding and gene number?
Table 5, tittle: “Summary statistics for protein-coding gene prediction”. Foot note: add bp; the symbol * is refer to two software??? It is not clear. One software a* and the other a + or something like that.
Table 6, tittle: “Summary statistics for functional annotation”. What is the meaning of the star??
Table 7, tittle: “Summary statistics for orthogroups analysis between annotations of Tieton and Satonishiki”. Change the tittle based in my former commentary. Also need a foot note explaining the meaning of NCBI and original (give the references).
Figure 1, tittle: “Pseudomolecule construction of sweet cherry by assigning scaffolds to seven genetic maps KS, CP, CK, AC, VG, JWF, and JWF1 are the genetic maps reported in Calle et al. 2018; Guajardo et al. 2015 ; Klagges et al. 2013 ; Peace et al. 2012 ; Shirasawa et al. 2017 ; Wang et al. 2015” . This tittle has to be changed!!! Also the order of the publications and the names should be the same.
Figure 2, tittle: “Species tree and gene family expansion analysis. A species tree were also constructed by using STRIDE , as part of OrthoFinder. A comparison was conducted using CAFÉ (version 4.2) . Compared with other plant genomes, 1,012 gene families had expanded and 3,642 gene families had contracted in sweet cherry genome”. Has to be changed!!! It is confuse. What is the meaning of the colors in the figure??
Figure 3, tittle: “Synteny analysis between the Tieton genome and Satonishiki genome Left is the draft assembly comparison and right is pseudomolecules comparison. X- and Y axes are sequences of Tieton and Satonishiki. Sequence similarity is indicated by colors”. Also need to be changed based in my former comments. Which similarity of the colors?? Nothing is explained in the text!!
It will be more informative if the authors discuss more each figure and table through the manuscript. At the moment is one phrase and in parenthesis the number of the figure and/or table.
The same arguments are valid for the supplementary data. The authors do not use that through the manuscript; only they cite them but no discussion or presentation of the data. This need to be improved, otherwise better not to add supplementary data that cannot be understood. In my former review I corrected some of them, for example the names of species always in italic. Table S2 still the same!!

Experimental design

'no comment'

Validity of the findings

'no comment'

Additional comments

Still, some minor issues with the English. For example lane 168: “OrthoFinder (version 2.2.7) was used to identify the orthologous genes from the thirteen plant genomes (Emms & Kelly, 2015) of the sweet cherry cv. ………”. Whys is that reference there? That reference that no describe 13 genomes!!! I will recommend to the authors to include the 13 references in the reference section of the manuscript.

---

## Author Rebuttal · Round 0.2

Jan 14, 2020

Dear Robert VanBuren,

Thank you for handling our revision of the manuscript previously entitled "A new *de novo* assembly of sweet cherry (*Prunus avium*) improves genome coverage and completeness". Reviewers' comments are very valuable and helpful for improving the quality and readability of the manuscript. Following their suggestions, we corrected and revised the manuscript. Please find our detailed responses (regular font) to the reviewers' comments (blue font in underlined) below and modified paragraphs in the paper (tracked changes).

For your comments:
- 1. More details are needed on the methods for assembling and anchoring the genome.

    **Response:** We appreciate your comments. More details were added to describe the methods for assembling (Line: 114-118) and anchoring the genome (Line: 125-139).

- 2. It would be useful to compare the reference genome presented here to the first sweet cherry genome sequence reported by Shirasawa et al., 2017. Simple comparative genomics analyses and a comparison of gene content differences would help strengthen this manuscript and validate the findings.

    **Response:** Two paragraphs have been added to analyze the whole sequence synteny and gene content difference between Tieton and Shirasawa genome assemblies and annotations (Line: 180-189, 279-296). Figure 3 and Table 7 were added in the revised manuscript.

- 3. There are a number of grammatical issues and the manuscript needs some heavy editing.

    **Response:** The revised manuscript was carefully edited by the PeerJ language editing service department.

For Reviewer 1
- Basic reporting
  This paper reports the whole-genome sequence of sweet cherry, for which the authors used 10X Genomics Chromium technology.
  Since the assembly presented in this paper is not dramatically improved in compare to the previous report, I recommend to delete "improves genome coverage and completeness" from the title.

**Response:** Thanks for your suggestion. We changed the title to "A *de novo* assembly of the sweet cherry (*Prunus avium* cv. Tieton) genome using linked-read sequencing technology"

➢ Experimental design
The authors used 10X Genomics Chromium reads to estimate the size of the sweet cherry genome (299.17 Mb) and to evaluate the assembly quality (99.02%). However, as the authors have already pointed out, it might be due to bias in the Chromium library missing 38 Mb. I strongly recommend the authors to analyze again with whole-genome shotgun reads obtained from a PCR-free method, e.g., TruSeq DNA PCR-Free Sample.

**Response:** We re-analyzed the genome size of Tieton based on 37-nt k-mer length rather than 17-nt and got the estimation of 341.38 Mb, which is very close to the genome size of 338 Mb estimated from the flow cytometry. The genome size, heterozygosity and repeat content values of Tieton genome were corrected in Line: 201-206.

➢ Validity of the findings
The text is totally descriptive and lacks any insights into biological aspects in sweet cherry.

**Response:** Our manuscript focuses on the sequencing method and assembly of sweet cherry genome using linked reads technology. To our knowledge, this is the first report of a genome assembly of *Prunus* plant using the 10X Genomics Chromium technology. We tried our best to improve the biological aspects in sweet cherry in the entire manuscript.

➢ Comments for the Author
Please consider the comments to improve this manuscript.

**Response:** Thanks for your valuable and helpful comments. We improved the the quality and readability of the manuscript following your comments.

For Reviewer 2
➢ Basic reporting
The paper: A new de novo assembly of sweet cherry (Prunus avium) improves genome coverage and completeness by the authors: Jiawei Wang, Weizhen Liu, Dongzi Zhu, Xiang Zhou, Po Hong, Hongjun Zhao, Yue Tan, Xin Chen, Xiaojuan Zong, Li Xu, Lisi Zhang, Hairong Wei and Qingzhong Liu prsent data for a new sweet cherry sequencing using the "Tieton" variety. I belive that the authors need to improve the introduction of the manuscript. There are few paper that should be considered, for instance: 1) Prunus genetics and applications after denovo genome

sequencing: achievements and prospects by Aranzana et al. Horticulture Research ( 2019)6 :58; Also is important to add info about organel sequencing: The complete mitochondrial genome sequence of sweet cherry (Prunus avium cv. 'summit') by Yang et aL., MITOCHONDRIAL DNA PART B 2019, VOL. 4, NO. 1, 1996–1997 AMONG OTHERS. The figures and tables need to be adreess in a more wide aspect.

**Response:** We followed the reviewer's suggestion to improve the introduction of the manuscript in Line 61-70. The two references were added, and figures and tables were improved.

➢ Experimental design
The paper present a new sweet cherry genome sequencing using two powerful tools that allow to improve the sequencing, however no much information is given for those methods: linked-read sequencing technology and the Supernova genome assembler. These techniques are very important for genome sequencing in plants and should be highlighted.
The research question is well defined but I stated previously it need to considerer more publication on the field.

**Response:** The linked-read sequencing technology and the Supernova genome assembler were highlighted in the introduction (Line:73-80), methods (Line:110-114), and results (Line: 210). In addition, we used the new title of the manuscript to highlight the linked reads sequencing technology.

➢ Validity of the findings
The results are good however they should discuss and compare their results with the previous sweet cherry sequencing (Shirasawa et al., 2017). This will allow to present more robust conclusions for this sequencing. Most of the genome sequencing papers in these do a deep genome sequencing with the same species or relates, for instance: peach; sweet cherry and apple.

**Response:** The sequence synteny and gene content difference were compared between Tieton and Shirasawa genome assemblies and gene annotations in Line 279-296.

Another concern is why the do not use the supplementary data through the manuscript?

**Response:** We carefully checked the order of supplementary data in the main text and used the supplementary data through the manuscript.

➢ Comments for the Author
The English need some improvement in the manuscript and supplementary data.

**Response:** We improved the English in the manuscript and supplementary data with the help provided by the PeerJ language editing service department.

Thank you very much for considering our manuscript for potential publication. I'm looking forward to hearing from you soon.

Sincerely,

Weizhen Liu, Ph.D.

Bioinformatics Laboratory,
School of Computer Science and Technology,
Wuhan University of Technology,
Wuhan, Hubei 430070 China
Email: liuweizhen@whut.edu.cn

---

## Round 0.3 · Minor Revisions

The authors have addressed the previous concerns raised by the reviewers, but there are still a few minor comments that need to be addressed before this manuscript is accepted.

1. The authors should release the genome assembly, annotation, and raw sequence files on public databases such as NCBI, Phytozome, CoGe, Genomic Database for Rosaceae (GRD), etc. This information should be summarized in a data availability section.

2. Each bioinformatics program used in this study should include version identification to ensure reproducibility.

---

## Round 0.4 · Minor Revisions

You are reporting a newly-assembled genome. To meet the PeerJ data sharing policy, you must deposit the raw reads **and** the assembled sequence in the appropriate databases (e.g. <https://www.ncbi.nlm.nih.gov/guide/howto/submit-sequence-data/>.

Since your raw reads are already in the NCBI SRA database, we suggest you use https://www.ncbi.nlm.nih.gov/genbank/eukaryotic_genome_submission/ however if you prefer, an alternative in China is <https://bigd.big.ac.cn/gwh/>

Another great alternative would be rosaceae.org:
<https://www.rosaceae.org/>

The files are available on Figshare, but they are difficult to find and submitting them to the appropriate database will help improve the impact and accessibility of this manuscript.

---

## Round 0.5 · accepted · Accept

Your manuscript is now suitable for publication.

---

## Author Rebuttal · Round 0.5

Apr 1, 2020

Dear Robert VanBuren,

We're thankful for your valuable comments on our manuscript entitled 'A de novo assembly of the sweet cherry *(Prunus avium* cv. Tieton) genome using linked-read sequencing technology'.

We've re-looked at the manuscript and changed in line with your comments. All of them have been accepted and incorporated into the revised version of this manuscript.

For your comments:

➢ 1. You are reporting a newly-assembled genome. To meet the PeerJ data sharing policy, you must deposit the raw reads **and** the assembled sequence in the appropriate databases. (e.g. <https://www.ncbi.nlm.nih.gov/guide/howto/submit-sequence-data/>.Since your raw reads are already in the NCBI SRA database, we suggest you use https://www.ncbi.nlm.nih.gov/genbank/eukaryotic_genome_submission/ however if you prefer, an alternative in China is https://bigd.big.ac.cn/gwh/.Another great alternative would be rosaceae.org: https://www.rosaceae.org/.The files are available on Figshare, but they are difficult to find and submitting them to the appropriate database will help improve the impact and accessibility of this manuscript.

**Response:** Accepted and the assembled sequence has deposited in NCBI under Accession number VTVB00000000. The section summarized the information was also changed in the manuscript.

Thank you very much for considering our manuscript for potential publication. I'm looking forward to hearing from you soon.

Sincerely yours

Weizhen Liu

Weizhen Liu, Ph.D.
Bioinformatics Laboratory,
School of Computer Science and Technology,
Wuhan University of Technology,
Wuhan, Hubei 430070 China
Email: liuweizhen@whut.edu.cn